# The Use of Blockchain Technology and OCR in E-Government for Document Management: Inbound Invoice Management as an Example

**Fatima Azzam** , **Mariam Jaber** , **Amany Saies** , **Tareq Kirresh** , **Ruba Awadallah** * , **Abdallah Karakra** , **Hafez Barghouthi** and **Saleh Amarneh**

Department of Computer Science, Birzeit University, Birzeit 71939, Palestine; akarakra@birzeit.edu (A.K.); hbargothi@birzeit.edu (H.B.); saleh.amarneh@gmail.com (S.A.)
* Correspondence: rawadallah@birzeit.edu

**Abstract:** The field of electronic government (e-government) is gaining prominence in contemporary society, as it has a significant influence on the wider populace within the context of a technologically advanced world. E-government makes use of information and communication technologies (ICTs) at various levels and domains within government agencies and the public sector. ICT reduces manual labour, potential fraud points, errors, and process lapses. The Internet's quick accessibility and the widespread adoption of modern technologies and disciplines, such as big data, the Internet of Things, machine learning, and artificial intelligence, have accelerated the need for e-government. However, these developments raise a number of data reliability and precision concerns. The adoption of blockchain technology by researchers demonstrates its efficacy in addressing such issues. The present study proposes the SECHash system model, which integrates blockchain and Optical Character Recognition (OCR) technologies for the purpose of regulating the processing of incoming documents by governmental agencies. As a case study to assess the proposed system paradigm, the study uses a document containing incoming invoices. The proposal seeks to maintain the integrity of document data by prohibiting its modification after acceptance. Additionally, SECHash guarantees that accepted documents will not be destroyed or lost. The analysis demonstrates that using the SECHash model system will decrease fraudulent transactions by eradicating manual labour and storing documents on a blockchain network.

**Keywords:** blockchain technology; data integrity; documents auditing; e-government; OCR

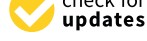



## 1. Introduction

E-government is a method of delivering government services through the use of information and communication technology (ICT) applications [1,2]. The concept of e-government refers to the development of information technology infrastructure within governmental institutions. Thus, facilitating immediate access to public data and enhancing public services [3]. Moreover, it offers various platforms and channels of communication for companies, agencies of government, as well as individuals, enabling their individual and collective interactions [4]. As a result, it is possible for individuals to acquire all notifications and documentation related to administrative procedures, with the exception of data that may have an impact on national security and confidential state matters. Therefore, there is a need for increased transparency and credibility in governmental transactions, as well as a greater assurance of safeguarding individual rights [2].

Furthermore, the implementation of e-government has been shown to efficiently reduce the expenditure of both time and financial resources in the assigned tasks, thereby contributing to the effective management of civil services and the promotion of economic growth [2]. In other words, labour and manual effort are the primary sources of gaps in the traditional government system, as any typo or misplacement of important paperwork can

put the owner at risk [5]. Hence, in conventional systems, it is imperative to put pressure on employees to concentrate on all aspects of their work. As per the given context, performing manual labour demands greater exertion and a larger physical area for managing records on paper [5].

Moreover, conventional systems provide an ideal environment for fraudulent activities and the act of forging. They permit duplicate data entry; there is no assurance of the integrity of the documents and the written data [5]. The implementation of an electronic system instead of the conventional paper-based system is expected to result in a reduction in the workforce, leading to a decrease in both mistakes in typing and security risks, as well as operating expenses [2].

Although e-government offers numerous benefits, there are still administrative challenges, such as fraud and bribery, that impede its widespread acceptance as an acceptable platform. For example, it is feasible to enter incorrect information into the database for a fee, despite the fact that this data is completely transparent. The authors of [6] indicate that a significant proportion, ranging from 35 to 40%, of applications submitted in the government sector are considered fraudulent. If corruption in government is not eradicated, it will undermine the rule of law and democratic values of nations. It will also have a negative impact on the economy of nations, impede the country's development, and provide individuals with inferior services [7]. Consequently, this will result in a decline of trust towards the governing authorities [8].

The objective of this study is to utilise blockchain technology in the realm of e-government, with the aim of capitalising on its efficacy [9]. It helps establish the groundwork for a new framework called SECHash (Scanning, Extraction, Confirmation, and Hashing of Documents). The process of validation involves the use of blockchain and optical character recognition (OCR) technologies to ascertain the genuineness of any incoming document. This procedure entails applying OCR to extract data from the submitted files and inserting the extracted data on the blockchain. The system stores all documents with cryptographic signatures and hashes. Therefore, it is impossible to transmit fake documents after the fact or use them after they have been stored. As a result, all data are unchangeable; once a document has been saved to the system, it cannot be deleted or altered.

Therefore, SECHash performs the following actions for the submitted document:

1. After confirmation, protect the document's data from any modification.
2. After acceptance, protect the document data from damage or loss.

The structure of this paper is as follows: Section 2 discusses related blockchain technology work and its contributions. Section 3 discusses the construction of the SECHash proposal system model. The concept demonstration and evaluation of the proposed system model are presented in Section 4. The concluding remarks, along with suggestions for future research, are presented in Section 5.

## 2. Related Work

This study relies on the utilisation of blockchain technology to augment the level of data security within the realm of e-government. Blockchain is a distributed digital ledger containing segments of cryptographically signed transactions in blocks. After undergoing confirmation and a consensus decision, each block is cryptographically connected to the preceding block (making it tamper-proof). As new blocks are inserted, it becomes increasingly difficult to modify older blocks, producing tamper resistance. New blocks are replicated across multiple instances of the distributed ledger within the structure of the network, and any disagreements are settled automatically using predetermined rules [10,11].

In the past few years, a substantial body of literature has developed around the idea of blockchain; its popularity is primarily attributable to the following features:

- Decentralisation: Transactions in systems that operate on separate or clustered servers are centralised and require the participation of a third party. The third party is entrusted with sending requests to the system's server(s) and waiting for a response.

If a single server goes offline, the entire system will fail, creating a single point of failure [12]. Additionally, in centralised systems, all authorities are dependent on a single entity's decisions. Consequently, decisions will take longer. Using the peer-to-peer network, blockchain was able to resolve these issues. The e-government system is a prime example of a centralised system that must be improved to provide better and quicker services [13].

- Immutability: The blockchain employs cryptographic hashes that cannot be reverse-engineered. In other terms, the blockchain's transaction history is immutable, permanent, and unalterable [14]. Immutability is one of the primary benefits of blockchain features that could aid in the reduction in corruption in government.
- Transparency: Blockchain transactions are public, traceable, and available to anybody with access [15].

Blockchain technology has been effectively applied in a variety of settings [16]. Many governments throughout the world are launching blockchain-based projects and have altered, proposed, or revised legislation to accommodate this new technology in various situations. According to IBM, 70% of healthcare professionals believe the application of blockchain will have the greatest impact on the industry. As a result of this technology, clinical trial management will improve, regulatory compliance will be ensured, and a decentralised platform for sharing electronic health records will be created (EHR) [17]. Furthermore, the application of blockchain in the agri-food supply chain can assure comprehensive food traceability, support market prices, protect the rights of employees, reduce the influence of supply chain intermediaries, and provide incentive systems to stimulate the growth of environmentally friendly initiatives [18,19]. Additionally, numerous academic studies on the use of blockchain technology in energy commerce [20,21], demand response programmes for microgrids [22,23], as well as peer-to-peer networks have been published [24,25]. Therefore, blockchain can facilitate the development of renewable energy sources [26]. Moreover, one of the most prominent applications of blockchain technology is protecting data privacy in various Internet of Things (IoT) domains, such as [27,28].

By achieving high levels of traceability, accountability, security, and high-quality services, governments employ such initiatives in an effort to earn the public's trust [29]. For instance, the U.S. Department of Homeland Security has adopted a number of Ethereum blockchain solutions in conjunction with distributed storage systems for data to avoid corruption and offer safe digital identity management [30]. OpenDChain is a blockchain-based application example developed in Spain [30]. This application intends to publish the datasets on the municipal website, which is accessible to the public. The Canadian government also hosts its blockchain explorer on the Interplanetary File System (IPFS) network, which enables users to seek for contribution and grant data [30].

Presently, various extant research papers demonstrate the significance of implementing blockchain technology. Fallucchi et al. [30] introduced an Ethereum blockchain-based framework for IPFS-based decentralised applications. This framework is employed for verification of documents without a requirement for a third party or a centralised organisation, with the assurance of its ownership and immutability. The implementation of this framework enables the organisation to disseminate its documentation, including financial reports, on the network with a high degree of transparency and at a low cost. Lacity and Van Hoek [31] demonstrate Walmart Canada's blockchain-based invoice processing solution. In this solution, the percentage of disputed invoices is reduced from 70 to 2%, and the processing duration is shortened from days and weeks to 24 h. As a consequence of this reduction, the cost goes down and the connection and communication are strengthened.

Rasool et al. [32] introduced DocsChain, an OCR and blockchain-based solution that enables the issuance and validation of degrees. OCR can capture the data on the degree and convert it to a machine-readable format. The documents were subjected to a hashing algorithm and the Proof of Existence (PoE) protocol was employed for the purpose of document retrieval [33]. Conversely, making use of physical documents may result in a compromise of their integrity. The authors of [34] proposed a solution for document

verification that employs an integration of blockchain, optical character recognition (OCR), digital signatures, and two-dimensional (2D) bar-code technologies. The experiment yielded a remarkable 100 percent accuracy. In the second stage of the trial, all methodologies were combined. To accomplish this, fresh pages were generated, the verification text was included, and the barcode at the bottom of the documents was appended. The integration of these technology combinations, along with the implementation of a validation process, facilitates the identification of any alterations made to documents.

Das et al. [35] proposed a framework for securing Architecture, Engineering, and Construction (AEC) projects. The first part of this framework is a smart contract utilising blockchain technology for the document certification process. The second part is a blockchain-based protected ledger data paradigm for auditing and controlling irrevocable documents (altered records). The third component is a modified Merkle–Patrick Trie (MPT)-based document version history data structure. In addition, Nizamuddin et al. [36] presented a solution for the online publication of digital content such as books. This framework's primary objective was to address the challenge of authorship and originality for online consumers. IPFS was used to store content, and blockchain smart contracts were used to monitor and manage content versions.

Alketbi et al. [37] proposed a novel blockchain paradigm that governments could use to develop a blockchain ecosystem for the provision of public services. The authors of this study have developed the blockchain model by conducting an inquiry into permissioned blockchain platforms and examining a use case for blockchain-enabled home rentals that has been implemented by the Dubai government. The outcomes of the suggested blockchain framework encompass a blockchain governance framework, a delineation of participants and their corresponding duties, and a network architectural template that specifies the diverse installation alternatives and their component elements. Additionally, the authors analyse the security and performance of the model, along with its lifecycle and blockchain services. The article additionally examines various instances of government usage of blockchain technology through the implementation of evidence of concept or prototypes. The study was founded on the design of Hyperledger Fabric, and the findings indicate the platform's pertinence to government services and use cases. The analysis of Hyperledger Fabric also entails the identification of the platform's actors, services, procedures, and data structure.

Páez et al. [38] presented a proposed framework for an electronic identification document (e-ID) system that utilises Blockchain technology and biometric authentication. The Electronic Identification (E-ID) system is utilised to authenticate individuals in various transactions such as notarial services, registration, tax declaration and payment transactions, basic health services, and recording business activities. The proposed method for validating transactions was developed utilising a blockchain framework to monitor and verify all transactions executed by individuals registered in the electoral registry. This approach guarantees security, consistency, expandability, traceability, and clarity. Furthermore, the study presents a blockchain network structure that is distributed and decentralised, encompassing all network nodes, databases, and government entities such as national registries and notarial offices.

Yavuz et al. [39] employed Ethereum wallets and the Solidity programming language to develop and evaluate a sample e-voting system as a smart contract for the Ethereum network. The feasibility of enabling Android platform users who do not possess Ethereum wallets to participate in voting is currently under discussion. The electoral data pertaining to ballots and votes shall be stored on the Ethereum blockchain subsequent to the conduct of an election. The collective agreement among Ethereum nodes is responsible for processing voting requests, which can be initiated by users via an Android device or directly from their Ethereum wallets. This agreement provides an open platform for electronic voting and investigates the dependability and efficacy of blockchain-based electronic voting systems in depth. Their application is limited to small-scale voting systems, so new problems may arise in larger systems.

Zhang et al. [40] suggested the theoretical basis and benefits of blockchain technology for managing and distributing government information. They also went into more detail about the blockchain-based solution. In addition, they developed a blockchain-based architecture for sharing government information and providing technical and administrative implementation techniques. The outcome was a framework for analysing government and user interaction with information. Moreover, this model of government information-sharing facilitates the provision of transparency in the distribution of government information. In addition, this initiative facilitates the development and enhancement of information sharing capabilities within the government, providing a standardised information-sharing framework for e-government across various departments and regions. The Table 1 below contrasts the above papers based on the maturity of their solutions and consensus algorithms.

**Table 1.** Comparing papers dedicated to blockchain.

| Paper | Scope | Maturity of Solution | | | | Consensus Algorithm | | | | | | |
|---|---|---|---|---|---|---|---|---|---|---|---|---|
| | | Experimental | Conceptual | Prototype | Proposal | Kafka Consensus | Proof-of-Stake | Proof-of-Luck | Tournament Consensus | Proof-of-Existence | Proof-of-Work | Proof-of-Integrity |
| (Fallucchi et al., 2020 [30]) | Security and Data Integrity | ○ | ○ | ● | ○ | ○ | ● | ○ | ○ | ○ | ○ | ○ |
| (Fallucchi et al., 2020 [30]) | Documents Traceability | ○ | ○ | ● | ○ | ○ | ● | ○ | ○ | ○ | ○ | ○ |
| (Fallucchi et al., 2020 [30]) | Documents Verification | ● | ○ | ○ | ○ | ○ | ● | ○ | ○ | ○ | ○ | ○ |
| (Lacity et al., 2020 [31]) | Invoice Processing | ● | ○ | ○ | ○ | ○ | ● | ○ | ○ | ○ | ○ | ○ |
| (Rasool et al., 2019 [32]) | Education (Degree Verification) | ○ | ○ | ● | ○ | ○ | ○ | ○ | ○ | ● | ○ | ○ |
| (Mthethwa et al., 2020 [34]) | Document Verification | ○ | ○ | ○ | ● | ○ | ○ | ○ | ○ | ● | ○ | ○ |
| (Das et al., 2022 [35]) | Document Version Management | ○ | ○ | ● | ○ | ○ | ○ | ○ | ○ | ○ | ○ | ● |
| (Nizamuddin et al., 2018 [36]) | Online Publication | ○ | ○ | ○ | ● | ○ | ● | ○ | ○ | ○ | ○ | ○ |
| (Alketbi et al., 2020 [37]) | Land Property Services | ○ | ● | ○ | ○ | ● | ○ | ○ | ○ | ○ | ○ | ○ |
| (Páez et al., 2020 [38]) | Authentication | ○ | ○ | ● | ○ | ○ | ○ | ● | ● | ○ | ○ | ○ |
| (Yavuz et al., 2018 [39]) | e-Voting | ○ | ○ | ● | ○ | ○ | ● | ○ | ○ | ○ | ○ | ○ |
| (Zhang et al., 2019 [40]) | Data Sharing | ○ | ● | ○ | ○ | ○ | ● | ○ | ○ | ○ | ○ | ○ |
| SECHash | e-Government | ○ | ○ | ● | ○ | ○ | ● | ○ | ○ | ○ | ● | ○ |

○ = Not supported; ● = Fully supported.

Despite the numerous benefits demonstrated by previous studies and research, all of them can fail or lose their advantages if implemented to a different field. This paper works on a radical and general solution to manage and store any incoming document in a blockchain network regardless of domain or business context. The amount of human intervention required to save the uploaded transaction document is minimised. Everything else is administered exclusively by the SECHash model. As a consequence of utilising the model, both human error and voluntary document modification are complicated processes. Moreover, because the documents are hashed, any modification requires new fragmentation calculations. Thus, our system architecture automates the data entry process with OCR technology and implements data integration with blockchain technology.

### 3. Proposed SECHash System Model

The proposed SECHash system model is based on blockchain and OCR technologies. This model will promote data correctness and integrity to governmental agencies for incoming documents. Furthermore, the model applies to any document management use case where the document is non-fungible, and its embedded data are helpful to digitise, as shown in Figure 1. As a case study, we implemented a proof-of-concept of this model specific to inbound/incoming invoice (or bill) management. This was performed due to the regular or semi-regular nature of invoices [41], as they have similar structures and are easier to perform OCR operations. Sections 3.1 and 3.2 define the SECHash model and show the system model workflow, respectively.

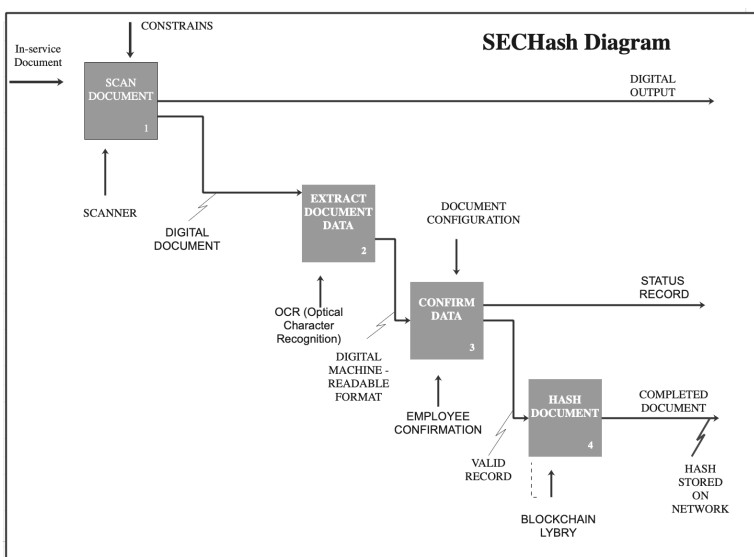

**Figure 1.** SECHash system model.

### 3.1. SECHash System Model Methodology

1.  SECHash System Design Process: The SECHash model is proposed to solve the problem of the lack of individuals' confidence in government due to corruption and fraud. In this paper, a case study in this area has been carried out. However, it will examine a specific use case related to external invoices/bills municipalities must pay. For example, suppose there is a maintenance bill from non-governmental organisations (NGOs) that the municipality must pay. The accountant or municipal employee will manually enter the bill data into the municipality's financial system. At this stage, the billing record is drafted. After auditing, the same employee or maybe another one confirms the draft bill, a journal entry is created on the financial system, and all financial statements are reflected in the accounts. One of the problems that may arrive from the manual entry of bills is that we cannot guarantee the correctness

of the bill data. Additionally, there is no way to verify the integrity of the entered information in the case of malicious actors.

The two main issues the SECHash model will look at are integrity and correctness. The suggested solution is to address these issues in a manner that is automated, transparent, and easy to understand in layperson's terms. Thus, promoting trust in municipalities and other similar governmental entities.

2. SECHash System Design Components: The proposal is based on three main components as follows:

    (a) OCR component: to capture the scanned bill's content, extract its data, and transform them into digital data that the municipal financial system can read.
    (b) Blockchain component: to hash the scanned bills documents and store the hashes in the network—this is a write-private network with an open-read permission model.
    (c) Bills explorer component: to allow the user to request and view the hash for any document.

3. Technology Selection: As a result of the recent literature review shown in Section 2, blockchain with OCR can be an excellent combination to address integrity and correctness issues. Therefore, the proposed SECHash system proposal is based on blockchain and OCR technologies in its solution.

4. Proof-of-Concept Implementation: this work builds the proposed SECHash to defend the idea. And to prove that blockchain technology in the governmental sector will preserve data integrity.

*3.2. SECHash System Model Workflow*

According to the SECHash model (see Figure 1), the document in our case study is the invoice that must be scanned at the beginning. Depending on the nature of the documents, there will be multiple restrictions on the scanned documents. For example, if the scanned copy is an invoice, the limits will be invoice ID and customer ID. In contrast, the output of the scanned function will be digital output (transaction records, checked date, etc.) and digitised documents, which will be sent to the OCR engine. Accordingly, the invoice/bill data are captured and extracted from the digital invoice document. In the confirmation and validation stage, the employee/user should confirm these data and the new status for the records will be reflected and stored in the database. Finally, the document and the good records will be hashed using the blockchain-based file-sharing and payment protocol (LBRY). As a result, these documents are stored in the network. Figure 2 shows the invoice/bill document workflow life cycle.

Figure 3 shows the state diagram of the invoice/bill document. In simple words, the state of the document is scanned at the beginning. Next, OCR captures the second state. Then, the state will be confirmed by the employee. Finally, the last state is hashed, meaning the bill document data are reflected in the accounting system and stored in the blockchain database.

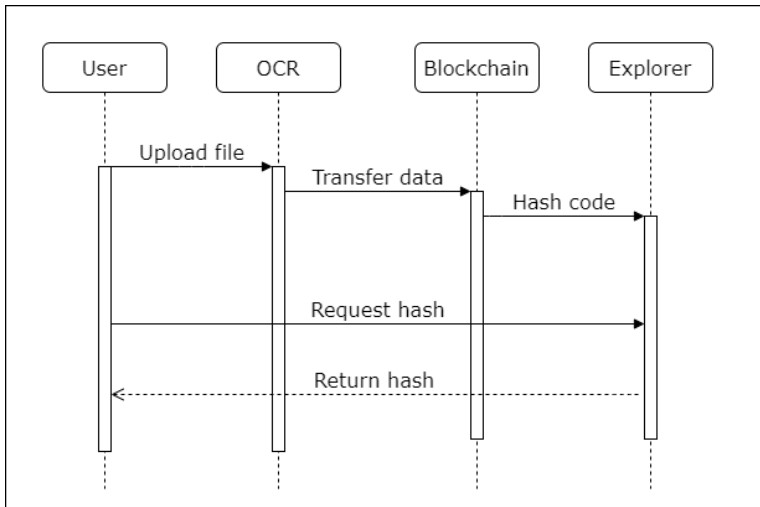

**Figure 2.** Sequence diagram for the system model.

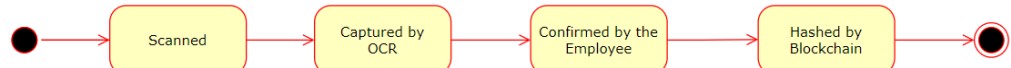

**Figure 3.** State diagram for bill document.

## 4. SECHash Proof-of-Concept

This section shows the technologies used to prove the concept of the SECHash proposal. It also explains the steps of how the proposal works and, finally, provides an evaluation.

### 4.1. Technology Selection

To implement the SECHash system model, the following technologies and components were selected considering several factors: the learning curve of the chosen technologies, local market, hardware, cost, performance, maintainability, stability, and security.

- *Django Web framework* [42]. It is an open-source web application framework written in Python. Because of its rapid development capabilities, Django is very sophisticated in today's market; it takes less time to build any application. It uses the Model View Template (MVT) design structure. Its name comes from the framework based on the model as the database, the view as the control function, and the templates as the user pages for communication interactions. A Django model acts as a database manager and uses two main commands:

  1. Django picks up the changes in the models.py file after python manage.py performs the migrations and sends the data to the PostgreSQL database, then migrate python manage.py. The Django system then saves all changes to the database system.
  2. Python manage.py runs the server at the end. This will start the project and give the user the local host address of the project running locally. The views.py file also handles project requests into template management API calls within demands. The user can describe her/his views using Python functions [42].

- *PostgreSQL database technology* [43]. It supports most SQL transactions and provides concurrency control. In addition, it offers modern features such as complex queries, triggers, views, transnational integrity, and allows adding data type extensions. It also provides functions, operators, and procedural languages. As a result, it is one of the world's most advanced open-source Database Management Systems (DBMS) [43].

- *LBRY* (https://spec.lbry.com/, accessed on 14 April 2023). A protocol enables a decentralised online content marketplace. Specifically, it uses blockchain to build a community controlled decentralised content platform. That lets users quickly publish,

host, search, access, download, and pay for content. LBRY introduces a new naming scheme that gives users complete control over the name of their content and uses blockchain to create a digital currency (LBC), a transparent distributed ledger, and a sync for all users. A global index of content metadata that also supports access to a unique namespace and supports new paradigms in digital content delivery [44]. The LBRY credits specifications are as follows:

- Max Supply: 1,083,202,000 LBC.
- Coin Type: PoW.
- PoW Algorithm: lbry.
- PoW Period: 20 years.
- Block Time: 2.5 min.
- Current Block Reward: 345 LBC.
- Premine: 400,000,000 LBC.

- *Hash function*. An algorithm takes user data as input and produces a fixed-length output (called a hash digest) for the input data. Federal Information Processing Standard (FIPS) 180–44 (Dworkin, 2015) [45] provided distinct requirements for hash functions algorithms certified by the National Institute of Standards and Technology (NIST). As a result, the Secure Hash Algorithms (SHA) family are the most commonly implemented hash function in different applications [46]. LBRY is a PoW algorithm that combines SHA-256, SHA-512, and RIPEMD hash functions. This algorithm enables fast and secure transactions on the LBRY network.

  1. SHA-256 has an output of 32 bytes and is displayed as a 64-character hexadecimal string. Therefore, it is considered secure and fast. The most common uses of SHA-256 are website authentication, digital signature, blockchain security, and file and fingerprint comparison in antivirus programs [47].
  2. SHA-512 is an algorithm based on non-linear functions that output 64 bytes. It is designed to prevent any cracking method and is unbreakable. At the same time, the user data are encrypted by hashing it to 128-bit hexadecimal characters. The SHA-512 algorithm is useful in many areas, such as internet security, digital certificates, and blockchain for secure password hashing [48].

- *Asprise* (https://asprise.com/royalty-free-library/python-ocr-api-overview.html, accessed on 5 March 2023). Since 1997, Asprise has provided customers worldwide with a wide range of Software Development Kits (SDKs) and APIs for programming libraries. Asprise OCR SDK is in high demand thanks to its high performance and royalty-free distribution model [49].

*4.2. SECHash Implementation*

The steps to implement a SECHash system model are as follows:

1. A Django project is created in the first step. Inside the `model.py` file, the following models are created:

   - The invoice model contains the following fields: invoice number, issued data, customer name, hash ID, total amount, currency, the image of the invoice document, address, email, phone number, notes, and foreign Key that has many records and data from other models for the invoice items.
   - The Invoice item model contains the following fields: product name, product description, quantity, the unit price for the product, and the amount.

2. The second step is migration, in which the SECHash system uses the `makemigrations` command. This command prepares all the SQL queries for the table creations, and the migration command executes the prepared queries to translate Python objects into tables in the PostgreSQL database. This is the role of the Object Relational Mapper (ORM).

3. Preparing the URLs as follows:

- Add new invoice document, invoice provided in Figure 4.
- Send invoice document to the blockchain as shown in Figure 5, which lists all the transactions that are sent to LBRY
- View the bill document from the blockchain database using the hash ID as shown in Figure 4. Figure 6 shows the claim ID of each document. Figures 5 and 7 show the transactions on the LBRY platform.

4. Implementing the logic for all the functionalities, which includes adding a new document, extracting invoice data from the uploaded document, creating a new record in the invoice table, and sending the document to the blockchain network.

## View image posted to LBRY

Detail view of 31dc977d1504a3561d2252f42344f39f1d55136d

### SELF EMPLOYED INVOICE

Invoice Number
IN-0012

### Billed to

| | |
|---|---|
| **Name** | Mark Smith |
| **Address** | 21th Street, Sad Village<br>Quezon City, Metro Manila, 1110 |
| **Email** | marks@example.com |
| **Phone Number** | (123) 456-7890 |

| | |
|---|---|
| **Date Issued** | 06/11/2020 |
| **Notes** | Bank Transfer |

**Work Description**

| | Description | Quantity | Unit Price($) | Amount($) |
|---|---|---|---|---|
| Work 1 | New Account Transfer | 1 | $70 | $70 |
| Work 2 | Premium Advantages | 2 | $30 | $60 |
| Work 3 | New Customer Credit | 1 | $100 | $100 |
| Work 4 | Account Credit | 2 | $20 | $40 |

**Total Amount** $270

**Figure 4.** View invoice image on the web using its claim ID.

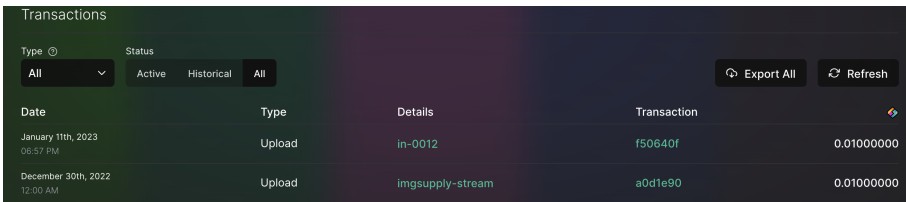

**Figure 5.** Transaction list.

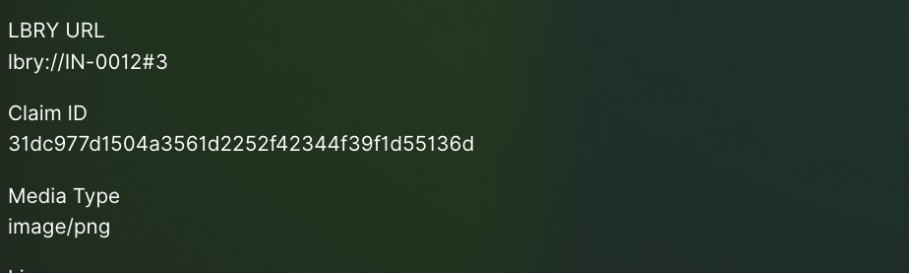

**Figure 6.** LBRY claim ID.

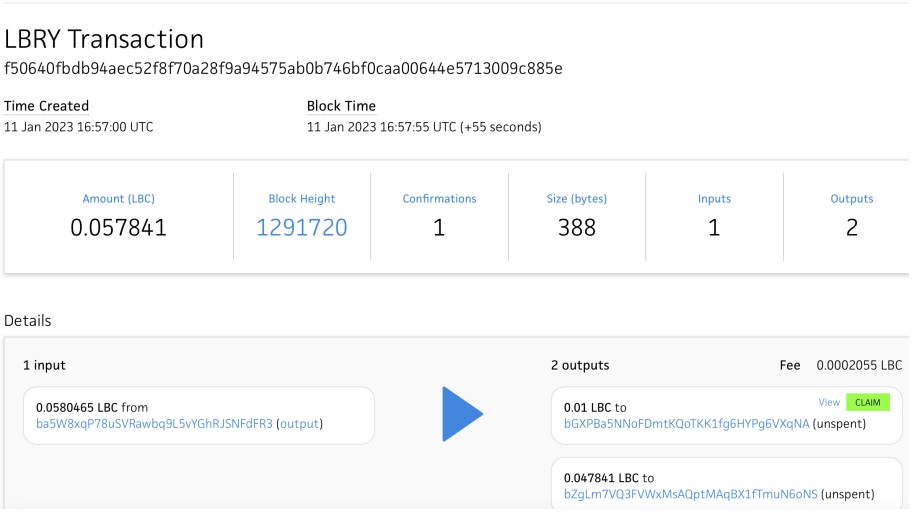

**Figure 7.** LBRY transaction. ID.

### 4.3. SECHach Evaluation and Discussion

This subsection evaluates the proposed SECHash model system and summarises the differences in results between the presented papers and the SECHash model.

#### 4.3.1. Storage Analysis

SECHash solves a significant problem in e-governance; central data storage capacity requirements and the resulting recourse to third parties may need to be more trustworthy. Utilising blockchain technology saves central storage space and ensures non-duplicate document copies for each transaction [50]. SECHash's data storage system is explicitly based on the LBRY blockchain. Thus, the government will trade in the LBRY blockchain network and create transactions that include copies of accepted documents, as mentioned in Section 4.2. The cost of storing data on the LBRY blockchain can vary depending on several factors, including the amount of data and current market conditions. The prevailing market price of LBC determines the cost of data storage, which today is estimated to be USD 0.010057 (as of 3 July 2023, https://coinmarketfees.com/lbry-credit, accessed on 20 May 2023). In addition, the fees charged for processing transactions and including them in the network will be added to the cost of trading. The price of fees varies according to network conditions. During times of heavy network traffic, fees may be higher. SECHash

does not focus on the speed of embedding the transaction into the network. Therefore, Shang et al. [51] recommended waiting for network conditions to improve, as fees are lower when the network is less congested.

In contrast, all papers offered in document data filing rely on centralised storage capacity or contracting with a third party. This process can be efficient. However, central storage would be a significant financial burden on government documents, estimated at USD 7–12 (https://dgtlinfra.com/how-much-does-it-cost-to-build-a-data-center/, accessed on 24 May 2023) million per megawatt of IT load. Moreover, data storage consumption is between 20–50 megawatts per year based on a 2023 study (https://www.aflhyperscale.com/articles/what-makes-hyperscale-hyperscale/, accessed on 26 May 2023). A government relying on centralised data storage would cost at least USD 140 million annually. As for the proposals outsourced to a third party, the third party must have an excellent reputation to secure sensitive government documents. Yet, it is possible to penetrate this third party either internally or from an external party without the knowledge of the government [52].

### 4.3.2. Data Security Analysis

SECHash achieves data integrity and authority based on the security of blockchain technology. The security of the SECHash system is based on the LBRY blockchain [53]. Data integrity is achieved by leveraging the technologies and concepts used in the LBRY. When documents are recorded in the blockchain network, changing or modifying them is difficult. This is due to the consensus mechanism in approving and confirming each transaction within each new block. Thus, any tiny change in the approved documents needs massive power of calculations. At the same time, it is easy to validate every document saved on the network using hashing algorithms. SECHash also achieves data authority. Since all the documents are hashed, and the data are encrypted, only authorised users can access these documents. In addition, using OCR technology will reduce the potential risk of human errors and data entry. Furthermore, the security features of SECHash are unaffected in the application domain.

In comparison, systems were proposed in previous studies and attempted to be applied to be effective in a specific field. At the same time, these systems lose their effectiveness and ability to save documents as required if used in other areas.

## 5. Conclusions

This paper proposes a new system model called SECHash for using blockchain technology in electronic government (e-government). Firstly, the paper reviews the most recent industry documents and the latest literature on blockchain technology and e-government and their uses together. Then, a new model of the system was introduced to deal with public information extraction in governments. This system model was implemented as a proof-of-concept using LBRY as the supporting blockchain, Django as the web framework, PostgreSQL as the metadata and application data database, and Aspire for OCR. In addition, it has been applied explicitly to manage incoming bills/invoices in municipalities. SECHash system model proves its efficiency in helping governments with digital transformation projects. It also demonstrates that it is a good solution for saving storage space. The SECHash system model will, therefore, result in significant cost reductions. SECHach is easy to confirm the validity of a document, while it requires a lot of computation power to modify a copy once registered in the blockchain network. Moreover, adopting OCR technology will reduce the chance of human error and incorrect data entry.

On the other hand, our approach has some limitations, including the lack of complex data or context on inbound invoice management accuracy and general budget reconciliation issues in government.

In the future, we aim to apply this system in a local context on the ground. We also seek to measure the impact of this system on individuals' confidence in the specific use case, including budget reconciliation (corruption/theft) and overall accuracy in handling incoming invoices (bills) to municipal organisations. Moreover, we intend to improve the

system model and implementation to use a private blockchain network instead of a public LBRY. We will also strive to fully automate the process by enhancing invoice data detection and extraction. Finally, we seek to implement a general purpose OCR extraction model to apply our generalised model across multiple domains.

**Author Contributions:** Conceptualization, T.K.; Methodology, F.A., M.J., R.A. and A.K.; Software, F.A.; Validation, T.K.; Formal analysis, F.A., A.S. and H.B.; Investigation, A.K. and S.A.; Resources, A.S., T.K. and R.A.; Data curation, M.J., A.S. and H.B.; Writing—original draft, F.A., M.J. and A.S.; Writing—review & editing, R.A., A.K. and S.A.; Visualization, M.J.; Supervision, R.A. and A.K.; Project administration, R.A.; Funding acquisition, R.A., A.K., H.B. and S.A. All authors have read and agreed to the published version of the manuscript.

**Funding:** This research was funded by Birzeit University.

**Institutional Review Board Statement:** Not applicable.

**Informed Consent Statement:** Not applicable.

**Data Availability Statement:** Data sharing not applicable.

**Conflicts of Interest:** The authors declare no conflict of interest.

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
