# Peer review of "The Use of Blockchain Technology and OCR in E-Government for Document Management: Inbound Invoice Management as an Example"

_applsci, doi:10.3390/app13148463_

Round 1

Reviewer 1 Report

This paper presents The Use of Blockchain Technology and OCR in E-Government

for Document Management: Inbound Invoice Management as an

Example, which are based on the all areas of Blockchain, this paper is directly related to the theme of this journal.

Overall, the paper is organized properly; So, the paper is accepted after following minor changes:

1.       Contribution of paper must be given in bullets in introduction section.

2.       Heading 2 state of art must be renamed with Related work

3.       Figure 1 is distorted so need to keep high quality images

4.       On page 9 hash code example is given so better to remove with Pseudo code

5.       Paper contains few grammar mistakes which will be cooperated in final version.

6.       Only 36 references are used which are very less number of references. It’s better to increase references up to 50.

7.       add few references related to blockchain, which are mentioned below

Ayub Khan, Reem Alkanhel, Hela Elmannai, and Sami Bourouis. "Lightweight-BIoV: Blockchain Distributed Ledger Technology (BDLT) for Internet of Vehicles (IoVs)." Electronics 12, no. 3 (2023): 677.

Kaishan Wu, Rashid Ali Laghari, Mureed Ali. "A Review and State of Art of Internet of Things (IoT)." Archives of Computational Methods in Engineering (2021): 1-19.

Minor english changes required some typo mistakes and grammar errors 

Author Response

Dear reviewer 
Thanks for the valuable comments.
We have considered all comments and applied them all to improve our paper. 
Please find the point-to-point response file in the attachment. 

Best regards.

Reviewer 2 Report

The authors of this study aim to preserve data integrity by using the potential of blockchain technology. 

The topic of the paper is interesting and in line with the scope of the journal. However, I think that some changes are necessary to improve its quality. They are listed below:

- In the abstract, the goal of the paper is not very clear. I would suggest to specify better the research gap you aim to fill and then your innovative contribution. 

- In the state-of-the-art section, I would suggest to include a paragraph where you talk about blockchain in general. Since it is a quite new technology, it is important to convince the reader about its strenghts. In particular, I would suggest to state that this technology was so far successful in various sectors. In this context, I would recommend to cite [R1], [R2] and [R3] with reference to healthcare sector, agri-food sector, energy sector, respectively.

[R1] Hasselgren et al. (2020). Blockchain in healthcare and health sciences—A scoping review. International Journal of Medical Informatics, 134, 104040.

[R2] Mirabelli et al. (2021). Blockchain-based solutions for agri-food supply chains: A survey. International Journal of Simulation and Process Modelling, 17(1), 1-15.

[R3] Wang et al. (2020). Integrating blockchain technology into the energy sector—from theory of blockchain to research and application of energy blockchain. Computer Science Review, 37, 100275.

- Table 1: I would suggest to include also your paper in this table, in order to understand better the differences with the existing scientific literature.

- Please, add more details about the generality of your work. Referring to the proposed approach, can be re-used easily in other contexts. Basically, I invite you to the describe better the meaning of this sentence: "However, a general-purpose OCR system is not in the scope of this research paper."

- Figure 1 should be better explained within the text. Please, add more details because the connections between the different blocks are not very clear. 

- I would suggest to move the information about the future works in the conclusions.

- In the conclusions, I do not think that the term "delinquent" is appropriate. Please, change it. 

- In the conclusions, please describe better the limitations of your study. 

I think that the English language is ok. However, please re-read the entire paper to correct any typos. 

Author Response

Dear reviewer

Thanks for your constructive comments
Please see the attachment point-to-point response file. 

Best regards, 

Reviewer 3 Report

In this paper, the authors propose SECHash, a framework that combines blockchain and OCR technology to manage incoming documents to government agencies while preserving data integrity.

The topic considered by the authors is extremely important in the current technological scenario. The proposed solution appears interesting although several insights are needed. In particular, the authors deal little with the issue of privacy, propose an exaggeratedly detailed description of the implemented system (e.g., providing UML diagrams and pieces of code) while they do not provide some scientific explanations to understand the actual innovations brought by the proposed system. To try to remedy these problems, I have the following suggestions for authors:

- Enhance related work by including related approaches to the one proposed, which address the issue of privacy in blockchain. For example, authors should consider the papers "A Privacy-Preserving Approach to Prevent Feature Disclosure in an IoT Scenario," "A two-tier blockchain framework to increase protection and autonomy of smart objects in the IoT," and several other similar papers. Probably the approaches described in these papers are orthogonal to their own and could be integrated into their system to address the privacy problem.

- Add a "Discussion" section in which they highlight the similarities, differences, novelties, strengths, and weaknesses of their approach compared to those already proposed in the literature.

The English is good. 

Author Response

Dear reviewer 

Thank you for your valuable comments, and we hope that the paper will satisfy you after the modification.
Please find the attachments.

Round 2

Reviewer 3 Report

The authors have complied with my requests.

There are only two citation errors; in particular in [27] the correct citation is:

Nicolazzo, S.; Nocera, A., Ursino, D.; Virgili L. A privacy-preserving approach to prevent feature disclosure in an IoT 456 scenario. Future Generation Computer Systems 2020, 105, 502–519. 457 28.

In [28] the correct citation is:

Corradini, E.; Nicolazzo, S.; Nocera A.; Ursino D.; Virgili L. A two-tier Blockchain framework to increase 458 protection and autonomy of smart objects in the IoT. Future Generation Computer Systems 2022, 181, 338–356

Author Response

Dear reviewer 
Thank you for noticing that.
The two references have been changed as requested.